# Phase Alignment of an Array Optical Telescope System Using Balanced Detection

**DOI:** 10.3390/mi14020409

**Published:** 2023-02-09

**Authors:** Yang Li, Qiang Wang, Yongmei Huang, Rongqi Ma

**Affiliations:** 1Key Laboratory of Optical Engineering, Chinese Academy of Sciences, Chengdu 610200, China; 2Institute of Optics and Electronics, Chinese Academy of Sciences, Chengdu 610200, China; 3School of Electronic, Electrical and Communication Engineering, University of Chinese Academy of Sciences, Beijing 100049, China

**Keywords:** phase alignment, balanced detector, an array optical telescope system, differential phase shift keying, the improved Mach Zehnder Interferometer

## Abstract

Differential phase shift keying (DPSK) modulation and multi-aperture receiving are effective means for suppressing flickering, deviation, and fragmentation of the light spot by atmospheric turbulence. What is challenging in coherent beam combination of such an array receiver system is to detect and compensate for phase deviation of sub-apertures. In this paper, a method of phase alignment of an array optical telescope system using balanced detection was proposed and demonstrated. The improved Mach Zehnder Interferometer (MZI) can demodulate the digital signal and recover the phase difference at the same time. It also brings a 3 dB gain to the receiver and improves the detection sensitivity of the system. Adequate simulations with OptiSystem and MATLAB were carried out to show that the power value remains near the ideal state of 2.75 mW, and the bit error rate is less than 10^−9^ after phase compensation, which indicates the effectiveness and accuracy of the proposed method. Furthermore, taking the communication interruption difference of ninety degrees as an example, the system bit error rate was reduced from 1 to 10^−35^, and communication was established again.

## 1. Introduction

Satellite-to-ground laser communication is the central link connecting the satellite laser communication network and the ground-based laser network. Wavefront distortion and intensity flicker appear on optical signals because the atmospheric turbulence constantly changes in near-earth space, which reduce the optical power of signals and the reliability of communication links [1,2]. Therefore, it has long been an urgent issue to overcome the effects of atmospheric turbulence in the field of free-space optical communication. In research, effective solutions have been proposed from two aspects of signal modulation and ground-based receiving.

To be more specific, for modulation detection technology. Gregory et al. proposed a method to suppress intrinsic noise by using symmetric double detectors. The experimental data show that the signal-to-noise ratio approaches the quantum limit with the common-mode noise suppressed [3]. Xiaoping Ma et al. proposed that the turbulence effect can be well suppressed in binary phase shift keying (BPSK) and differential phase shift keying (DPSK) modulation [4,5]. DPSK stores the phase information in the relative phase of adjacent symbols. Phase fluctuation of neighboring symbols is determined by the same atmospheric turbulence, and the change between them is consistent. Therefore, the bit error rate (BER) caused by inter-symbol interference is reduced, and the transmission performance is better. Moreover, DPSK demodulation does not need local oscillator light and frequency locking, it only needs to delay part of the signal and recover the digital signal through delay interference. Therefore, it also has a simpler detection structure. In 2019, the laser communication relay demonstration (LCRD) and verification test finally proved to be a success in the USA. It adopted DPSK and pulse position modulation (PPM)-compatible communication systems for the first time and achieved long-distance satellite-to-ground laser communication with a communication rate of 2.88 Gps and a communication distance of 45,000 km [6,7,8]. It goes a step further in NASA’s Integrated LCRD Low-Earth Orbit User Modem and Amplifier Terminal (ILLUMA-T) project [8].Furthermore, the High-speed Communication with Advanced Laser Instrument(HICALI) project has been initiated by the National Institute of Information and Communications Technology (NICT), aiming for a 10 Gbps-class laser link using DPSK [9]. For the receiving system, large aperture telescopes are restricted by high cost and complex structures, such as the Shack–Hartman sensor, variant mirror, fast mirror, and other devices [10,11]. Reducing the receiving aperture of the telescope is the simplest way to overcome atmospheric turbulence. The United States completed the lunar laser communication demonstration system (LLCD) test in 2013 [12]. The test verified the equivalent relationship between multi-aperture and large aperture [13]. Compared with a large aperture, the multi-aperture receiving system has the advantages of lower cost, improved diffraction limit performance, smaller gravitational effect, scalability, and so on [14,15].

In this work, DPSK modulation and multi-aperture receiving technology were adopted to improve the performance of the communication system. However, the tilt of optical axes brings uncertain phase deviation. Studying the phase deviation of sub-apertures is a breakthrough direction to ensure the quality of high-resolution imaging and space optical communication. The novelty of this paper is the innovated Mach Zehnder Interferometer (MZI) and the establishment of the autodyne detection mode, which replaced one optical signal and one zero position light with two optical signals. It brought a 3 dB gain to the demodulation input so that the sensitivity of the system was improved, and the transmission performance was optimized. Further, differently from existing methods, which implemented DPSK demodulation and optical signal phase difference detection separately using different technologies [16,17,18], this study combined the two. Specifically, we finished the identification by using the voltage signal from the differential output port of the balance detector after we had connected the demodulator and the balance detector. The sign was used to demodulate the digital signal and the value was used to calculate the phase difference. This paper is organized as follows. The design and the algorithm principle of phase compensation for an array optical communication telescope is presented in Section 2. The simulation experiment process of digital signal recovery and phase difference detection is exhibited in Section 3. The results and results are described in Section 4. Additionally, Section 5 summarizes this paper.

## 2. Theories and Design

### 2.1. The Structure of Phase Compensation

To solve the phase deviation caused by the tilt of the optical axis of the multi-aperture receiving system, the structure of MZI was innovated, which is a high-precision detection method with a simple structure proposed. An array optical communication system based on balanced coherent detection was designed to evaluate the communication quality before and after phase compensation. It is shown in Figure 1.

The transmitter generates CSRZ-DPSK optical signal by LiNbO3 MZM. The signal was transmitted to atmospheric space through a collimator with a diameter of 80 cm and received by a multi-aperture system. The diameter of both receiving sub-apertures were 40 cm, and the spatial optical signal was coupled into two single-mode fibers. Then, the signal was transferred to the balance detector, and the phase error was compensated by a phase shifter, to reduce the error rate and ensure the accurate demodulation of the digital signal. The key technology of the system is to integrate the phase deviation of the sub-aperture into the adjacent symbols of the DPSK demodulation, and to improve the sensitivity of the system by using the autodyne balance detection. The difficulty lies in the construction of innovative MZI to meet the input condition of the balanced detector, wherein the phase difference of the input optical signal from the same source is π.

### 2.2. Theories of the Structural Innovation

Mach Zehnder Interferometer (MZI) is a key device of DPSK demodulation. Figure 2 shows the schematic structure diagram of an MZI, which includes two cross-couplers and two optical fibers. As the inputs of MZI, the signal light and zero value light are equally split into two optical signals by the first cross-coupler. Then, the information flow of adjacent bits is obtained by setting a delay of one bit on the upper arm. Additionally, the phase information of the light is converted into the intensity information of the light with interference occurring at the output port of the second cross coupler. The interference output of MZI and the input of the balance detector are connected by optical fibers. Finally, the demodulation of the DPSK optical signal is completed by using the electrical signal.

This study improved the MZI and combined it with the balance detector to establish the structure of DPSK demodulation and phase detection. The difference is that the structure reduced the use of the front-end cross-coupler. Therefore, this design of the demodulator combines digital signal demodulation and phase difference detection. The structure diagram is shown in Figure 3.

Differently from direct detection, heterodyne detection is to input a channel of signal light and a channel of eigen light, and autodyne detection refers to the coherent detection of two optical signals from the same laser source [19]. Compared with Figure 2 and Figure 3, the innovative method used a phase shifter to replace the first coupler of MZI in the structure. In terms of input limit, the innovative structure directly inputs two channels of signal light, but MZI inputs one channel of the optical signal and one channel of the zero-value optical signal.

As shown in Figure 3, two optical signals were input to the one-bit delayer to complete the detection process of autodyne detection. Two forms of optical signals were obtained at the input port of the balanced detector. One was the interferometric phase length signal and the other was the interferometric phase extinction signal. $E_{11In}$ and $E_{12In}$ are the input ports of the receiver structure and $E_{11In}=E_{0},E_{12In}=E_{0}$, and the signals obtained by the phase adjustment of the quadrature phase shifter are $E_{11Out}$ and $E_{12Out}$, which are described as Equation (1).
(1)$$\left(\begin{array}{c} E_{11Out}\\ E_{12Out} \end{array}\right)=\left(\begin{array}{cc} 1 & 0\\ 0 & -j \end{array}\right)\left(\begin{array}{c} E_{11In}\\ E_{12In} \end{array}\right)=\left(\begin{array}{c} E_{0}\\ -jE_{0} \end{array}\right)$$

The duration of the delayer is strictly controlled to one bit period. The input optical signals of the coupler are obtained as $E_{21In}=E_{0},E_{22In}=-jE_{0}$, when adjacent symbols have the same phase.

For the optical coupler, the input ports are $E_{21In}$ and $E_{22In}$, and the output ports are $E_{con}$ and $E_{des}$. The transmission matrix for the cross is described in Equation (2) [20,21].
(2)$$\left(\begin{array}{c} E_{1Out}\\ E_{2Out} \end{array}\right)=\alpha \left(\begin{array}{cc} \sqrt{1-c} & pj\sqrt{c}\\ pj\sqrt{c} & \sqrt{1-c} \end{array}\right)\left(\begin{array}{c} E_{1In}\\ E_{2In} \end{array}\right)$$
where p is the case of conjugate with the value of +1 or −1; α is the additional loss with the ideal value of 0 on a logarithmic scale; and c is the coupling coefficient, and the value is 0.5 with the splitting rate of 50:50. Equation (3) describes the output signals of the cross couple in this case.
(3)$$\left(\begin{array}{c} E_{con}\\ E_{des} \end{array}\right)=\left(\begin{array}{cc} \frac{\sqrt{2}}{2} & j\frac{\sqrt{2}}{2}\\ j\frac{\sqrt{2}}{2} & \frac{\sqrt{2}}{2} \end{array}\right)\left(\begin{array}{c} E_{21In}\\ E_{22In} \end{array}\right)=\left(\begin{array}{c} \sqrt{2}E_{0}\\ 0 \end{array}\right)$$

Similarly, the input optical signals of the coupler are obtained as $E_{21In}=E_{0},E_{22In}=jE_{0}$, when the phase difference between adjacent symbols is π. In this case, the output signals of the cross couple are written as Equation (4).
(4)$$\left(\begin{array}{c} E_{con}\\ E_{des} \end{array}\right)=\left(\begin{array}{cc} \frac{\sqrt{2}}{2} & j\frac{\sqrt{2}}{2}\\ j\frac{\sqrt{2}}{2} & \frac{\sqrt{2}}{2} \end{array}\right)\left(\begin{array}{c} E_{21In}\\ E_{22In} \end{array}\right)=\left(\begin{array}{c} 0\\ j\sqrt{2}E_{0} \end{array}\right)$$

The amplitudes of the constructive interference and the destructive interference are given in Equations (5) and (6), respectively.
(5)$$E_{con}=A\left(e^{j\varphi_{n-1}}+e^{j\varphi_{n}}\right)$$
(6)$$E_{des}=A\left(e^{j\varphi_{n-1}}-e^{j\varphi_{n}}\right)$$

According to the mathematical analysis formula, the function of the external beam splitting structure of DPSK demodulation and phase detection is the same as the MZI, while the input optical signal at the receiving end is uniformly generated by the same light source. The voltage output of the balance detector can be expressed as Equations (5) and (6).

The first photodetector output is shown in Equation (7).
(7)$$u_{1}=i_{1}R=RBE_{con}{}^{2}=A^{2}RB{\left(e^{j\varphi_{n-1}}+e^{j\varphi_{n}}\right)}^{2}$$

The second photodetector output is shown in Equation (8).
(8)$$u_{2}=i_{2}R=RBE_{des}{}^{2}=A^{2}RB{\left(e^{j\varphi_{n-1}}-e^{j\varphi_{n}}\right)}^{2}$$
where A is the amplitude; R is the resistance value; B is equal to ηe/hν, which represents for the responsivity of the photodiode; η is the quantum efficiency of the photodiode; e is the amount of electron charge; h is the Planck constant; and ν is the frequency of light. Ideally, the value of B is 1 A/W.

As shown in Figure 3, the balanced detector is composed of two photodiodes with the same parameters. They work in the reverse voltage region, which converts electrical signals into optical signals and outputs them differentially. This output structure can well suppress common-mode noise and is conducive to coherent optical communication. The resulting differential voltage is described as Equation (9).
(9)$$u=A^{2}RB{\left(e^{j\varphi_{n-1}+\varphi_{miss}}+e^{j\varphi_{n}}\right)}^{2}-A^{2}RB{\left(e^{j\varphi_{n-1}+\varphi_{miss}}-e^{j\varphi_{n}}\right)}^{2}=2A^{2}RB\cos(\varphi_{miss}+\Delta \varphi )$$
where $\Delta \varphi =\varphi_{n-1}-\varphi_{n}$ represents the phase difference of adjacent symbols when there is no disturbance, and the value can only be 0 or π; and $\varphi_{miss}$ is the phase difference of two optical signals when considering disturbance. Equation (9) includes the relative phase required by DPSK demodulation and the phase error of the sub-aperture. A detailed derivation process will be demonstrated in the third section by utilizing the phase difference through electrical signal.

### 2.3. Design of the Algorithm

The derivation and calculation process of phase error and the demodulation of the optical signal is shown in Figure 4, which is the theoretical basis for phase detection in this paper.

As shown in Figure 4, $E_{11In}$ and $E_{12In}$ are, respectively, input to the external split DPSK demodulation and phase detection structure by two optical fibers to carry out high-precision detection of optical signal phase difference and digital signal recovery. The phase difference meets $\varphi_{12In}-\varphi_{11In}=\varphi_{miss}$.

Adding a fixed phase difference of +π/2 to the optical fiber arm where $E_{12In}$ is located. The two optical signals are represented as $E_{11Out}$ and $E_{12Out}$ after this stage, and there is $\varphi_{12Ou}{}_{t}-\varphi_{12In}=+\pi /2$, which equals $\varphi_{12Ou}{}_{t}-\varphi_{11Out}=\pi /2+\varphi_{miss}$. The outputs are $E_{21In}$ and $E_{22In}$ by adding the one-bit delayer to the optical fiber arm where $E_{11Out}$ is located. Bringing the phase difference of adjacent symbols, expressed as Δφ, and the value of Δφ can only be “0” or “π”, which is used to demodulate to get the binary “0” code or “1” code. The phase difference is obtained as $\varphi_{22In}-\varphi_{21In}=\pi /2+\Delta \varphi +\varphi_{miss}$.

Finally, we obtain $E_{con}$ and $E_{des}$ through a 3 dB coupler. The conjugation of the coupler used in this paper is p=1, The phase difference is obtained as $\varphi_{con}-\varphi_{des}=\pi +\Delta \varphi +\varphi_{miss}$. Therefore, the electrical signal of the differential output port of the balance detector is $i=2A^{2}B\cos(\Delta \varphi +\varphi_{miss})$; that is, $u=2A^{2}BR\cos(\Delta \varphi +\varphi_{miss})$.

The voltage signal displayed by the oscilloscope is the maximum value. Thus, the value of amplitude can be obtained when the phase difference is zero. The power value collected by the power meter is used as the compensation factor. The relationship is $P=U^{2}/2R$ between the power and voltage of the pure resistance circuit, and R takes 1 in the simulation. Therefore, the mathematical relationship with P is established as $\varphi_{miss}=\frac{1}{2}\arccos\left(\frac{P}{2A^{4}}-1\right)$.

## 3. Simulation Experiment Process

OptiSystem 15 (Ottawa, ON, Canada) is a practical optical simulation software with various optical component libraries and reasonable system classification, which is produced by Optiwave. The software has complete simulation functions for optical communication and a rich component library. Simulation data such as eye diagrams, waveform and power spectrum can be directly reflected in the corresponding test and analysis devices. It has been widely used in the field of optical communication. This work is shown in Figure 5.

### 3.1. Transmitter

Mach Zehnder Modulator (MZM) excels at modulating light intensity and phase because of a characteristic of wavelength independence. It is capable of high-data-rate modulation above 40 Gbps, which has become the basis for many advanced optical modulation formats [4]. Half-bit-rate NRZ-DPSK wave pattern was generated through the first modulator. The NRZ code enters the second modulator to obtain full-bit-rate RZ-DPSK by adding half-bit-rate clock signal (5 GHz). The process is shown in Figure 6.

According to the value of bias voltage $V_{bias1}$ and $V_{bias2}$, MZM usually works at three bias points: peak, zero and quadrature. Signals with different duty cycles are generated at different bias points [22,23]. This is shown in Table 1.

Compared with NRZ-DPSK and other RZ-DPSK modulation patterns, CSRZ-DPSK modulation pattern at the transmitting end, as RZ-DPSK signal with 67% duty cycle, has lower requirements on system signal-to-noise ratio(SNR) and better suppression effect on nonlinear effects [24,25]. Therefore, the CSRZ-DPSK signal is generated by cascading LiNbO_3_ MZM in this simulation. Finally, the output signal of the transmitter is observed by the observation instrument, and the basis is provided for the comparison and analysis of the receiver. Table 2 shows the parameter values.

Figure 5 and Table 2 show that the center wavelength was 1550 nm, which is widely used in Satellite-ground coherent communication systems. Furthermore, the optical fiber loss is the lowest to integrate with the ground optical fiber communication network simply when the working wavelength is 1550 nm [26]. Xiaoping Ma et al. verified that the turbulence tolerance is widened and the detection efficiency of the deviation exceeds 95% when the communication rate reaches 2.5 Gbps and 10 Gbps [18]. Additionally, Gang Wu proposed that when the transmission rate is 40 Gbps, the signal will be distorted due to the fiber characteristics such as nonlinear effect, dispersion and polarization mode effect [27]. Therefore, the bit rate was defined as 10 Gbps. The input terminal uses a pseudo-random binary sequence (PRBS) as a code containing user information. It was copied into three channels through the cross-module, and the first channel was kept in the format of binary sequence; the second channel was input to the NRZ pulse generator and was maintained in a rectangular pulse format; the third channel was used for DPSK modulation. The extraction process is carried on as follows:

Step 1. The binary sequence is inputted into the differential precoding system composed of a not gate, a delay and an exclusive OR gate. The conversion process of analytic code type is shown in Table 3.

Step 2. The modulated data is transmitted to the NRZ pulse generator to output the NRZ signal with digital information. The output optical power of the laser is set to 10 dBm. The NRZ signal and the laser carrier signal are sent to the first stage LiNbO3 MZM to generate the NRZ-DPSK signal. The NRZ-DPSK signal and the sinusoidal voltage signal are input to the two-stage MZM, and the frequency of the driving voltage is set to be one half of the data bit rate. CSRZ-DPSK signal is obtained corresponding to the null offset point of the LiNbO_3_ MZ transfer function, when the carrier frequency is 0.

Step 3. Observing the power spectrum and the waveform of the CSRZ-DPSK modulation signal. The energy of the CSRZ-DPSK modulated signal at the transmitting end was recorded as −21.857 dBm.

### 3.2. Receiver

The receiving end combines digital signal demodulation and phase difference detection in the same receiving module, and uses the output value of the detector as a feedback factor to compensate for the phase difference.

In order to simulate the phase difference caused by the tilt of the optical axis of the factor aperture, we add a phase error element in one arm. The phase shifter in the OptiSystem component library can only provide one input port and one output port. However, the phase shifter with single port input cannot be directly used. It needs to rely on MATLAB to complete the joint simulation, and MATLAB expands the electric input port for feedback adjustment [28]. Set phase error to generate phase error; set phase correct to correct the phase error. The binary sequence and NRZ pulse waveform are used as the DC signal generator, and the amplitude is set to one. The feedback signal is directly received from the low-pass filter at the receiving end. The parameter values are displayed in Table 4.

As shown in Table 4, the delay time of one bit is set in the time delay module, which is equivalent to 0.1 ns in this system. It is shown in Equation (10)
(10)$$Delay=\frac{1}{Bit\;rate}=0.1\;\mathrm{ns}$$

Then, 1 dB noise is added to the second coupler, and the amplitude is found to be 0.161 v with OptiSystem. The final received differential mode signal is electrically filtered, and the cutoff frequency of the low-pass filter is set to 8 Gbps. It is shown in Equation (11).
(11)Cutoff frequency=0.8∗Bit rate=8Gbps

The signal output by the low-pass filter is input to the third port of the bit error analyzer as the signal wave at the receiving end. A pseudo-random binary sequence (PRBS) of the first port receiving end of the bit error analyzer; the second port of the bit error analyzer receives the NRZ pulse sequence at the sending end.

### 3.3. Signal-to-Noise Ratio

In practice, the spectral ratio of the coupler inaccuracy, the shot noise and thermal noise are existed in the system. Additionally, two diodes of the balance detector are not exactly the same in the reality [29,30]. From Equations (7) and (8), the SNR of this communication system was derived in Equation (12).
(12)$$SNR=\frac{{\left(\frac{e}{hv}\right)}^{2}{\left(S_{1}\eta_{1}+S_{2}\eta_{2}\right)}^{2}A^{4}\varepsilon \left(1-\varepsilon \right)R}{\frac{e^{2}A^{2}}{hv}\left[S_{1}\eta_{1}\left(1-\varepsilon \right)-S_{2}\eta_{2}\varepsilon \right]\Delta fR+Var\left(n_{ex1}-n_{ex2}\right)+4k_{b}T\Delta fR}$$
where S is the photosensitive area of a diode; η represents the quantum efficiency of a diode; ε is the spectral ratio of the coupler; and Δf is the measured bandwidth. Thermal noise is $P_{T}$, which is equal to $4k_{b}T\Delta fR$, and $k_{b}=1.38\times 10^{-23}J/K$, which is the Boltzmann’s constant. Shot noise is expressed as $2eI_{DC}\Delta fR+Var\left(n_{ex1}-n_{ex2}\right)$, and $I_{DC}=\frac{e}{2hv}\left[S_{1}\eta_{1}\left(1-\varepsilon \right)A^{2}-S_{2}\eta_{2}\varepsilon A^{2}\right]$, $n_{ex}$ is excess noise.

Quantum efficiency has a significant impact on the SNR. SNR can be simplified into Equation (13) under the condition of the excess noise, and thermal noise can be ignored. The value of ε is 0.5. $S_{1}=S_{2}$ and $\eta_{1}=K\eta_{2}\left(0<K<1\right)$.
(13)$$SNR=\frac{\eta_{2}{\left(K+1\right)}^{2}A^{2}}{2hv\left(K-1\right)\Delta f}$$

Equation (13) shows that the SNR increases with the increases in the quantum efficiency, if the proportional coefficient of the quantum efficiency of $\eta_{2}$ remains the same. The SNR varies with *K*, while the conversion coefficient of $\eta_{2}$ is determined, which is shown in Figure 7, which indicates that the normalized signal-to-noise ratio is better than 94.8%, if K>0.97 holds.

## 4. Results and Discussion

On the basis of the above discussion on the subject scheme and simulation theory, we will analyze and interpret the simulation experimental results, including the power size, demodulation signal, and the result feedback of the eye diagram.

In the first step, the two optical signals received at the receiving end were kept free of other relative phases except for the one-bit phase difference generated by modulation and the π phase difference generated by detection. That is when the transmission process does not affect the two optical signals.

In the second step, phase interference was added to the transmission section to simulate the phase difference caused by the optical axis tilt. Additionally, the results of the simulation were recorded and analyzed.

Thirdly, according to the relationship between the voltage signal received by the oscilloscope and the phase difference between the two optical signals, the electrical signal feedback module was designed to correct the phase difference.

In this simulation, the bit error analyzer was used as a visual tool to evaluate the communication quality of the system. We obtained Q factors and eye diagrams as follows.

As shown in Figure 8 and Figure 9, ideally, the eye pattern was symmetrical and wide open with the upper and lower eyelids both thin. Therefore, there was a high noise capacity limit, small jitter, a bit error rate of 0, and a large Q factor reaching 369.391 to obtain excellent signal quality. When there was a phase difference of π/2, the eye-opening was significantly smaller, even zero, the upper and lower eyelids were wider, the jitter was increased, the bit error rate was one, the Q factor was zero, the communication quality was significantly decreased, and the communication could not be completed. After the phase compensation, the eye-opening was significantly larger, the upper and lower eyelids were thinner, the jitter was reduced, and the bit error rate was 3.42 × 10^−35^, which was 10^29^ orders of magnitude less than that when it was not adjusted. It can communicate normally, the Q factor was increased to 12.66, and the communication quality was significantly improved.

The power values of the three cases were recorded. Under ideal conditions, the power was 2.75 mW. Additionally, the power decreased to 3.30 × 10^−6^ W, when the phase difference of π/2 was added. It increased to 2.62 mW after phase correction, and the power was significantly increased.

The waveforms of the three cases and the waveforms of the modulation end are recorded as follows:

As shown in Figure 10 and Figure 11, when there was no phase difference, the demodulation waveform and modulation waveform was consistent. When the phase difference reached π/2, the waveform was very messy. The waveform shape was improved when the phase was corrected.

The analysis of various phase perturbations was carried out as follow, and the obtained power values were used to solve the relative phase error values. Feedback compensation was then carried out to meet the requirements of phase consistency, leaving the combined signal’s power values essentially unchanged and the BER optimized.

Power comparison diagrams before and after phase difference correction are shown in Figure 12. It is intuitively obvious that, before the phase difference correction, the power size gradually shrunk as the phase difference increased. The power size had a tendency to be stable following the phase difference correction, and its value was stable at the power value without the phase difference. 

Figure 13 depicts the BER before and after phase difference correction. Because the BER varied widely, we used the exponential power for statistical analysis, and we could see that the BER values were larger before the phase difference correction, and the BER decreased after the phase difference correction, and their powers were all less than −9, indicating that the BER was better than 10^−9^, ensuring normal communication.

The comparison of the Q parameter before and after the phase difference correction is shown in Figure 14. When the phase difference was zero, the value of the Q parameter was greater. The Q parameter decreases gradually as the phase difference value increased, eventually reaching 0 rad when the phase difference reached 1.57 rad or −1.57 rad. However, the figure shows that the Q parameters were all increased after the phase difference correction of this scheme, indicating that the communication quality was improved. 

## 5. Conclusions

In conclusion, this study proposed and demonstrated a method of phase alignment of an array optical telescope system using balanced detection. In this paper, the phase difference of the signal optical path was combined with the DPSK demodulation module. The electrical signal information of the differential output port of the balance detector was used to highly accurately solve the phase difference between the two optical signals. Specifically, the sign was used to demodulate the digital signal, and the value was used to calculate the phase difference.

The simulation results showed that the calculated phase difference information of the optical signal was compensated by feedback, so the corrected power value remained around 2.75 mW, the bit error rate was less than 10^−9^, and the Q parameter was improved. It is demonstrated that the scheme proposed in this paper can support signal combination, coupled signal power stability, and communication link reliability. As a result, the influence of the balance detector asymmetry can be ignored, if K>0.97 holds. Thus, a more reliable performance of the communication system can be achieved by utilizing the SNR data in engineering applications.

In addition, this paper verifies the detection of phase differences between two apertures, which can be extended to more aperture experiments using similar principles. Furthermore, atmospheric turbulence data can be used to improve the system.

## Figures and Tables

**Figure 1 micromachines-14-00409-f001:**
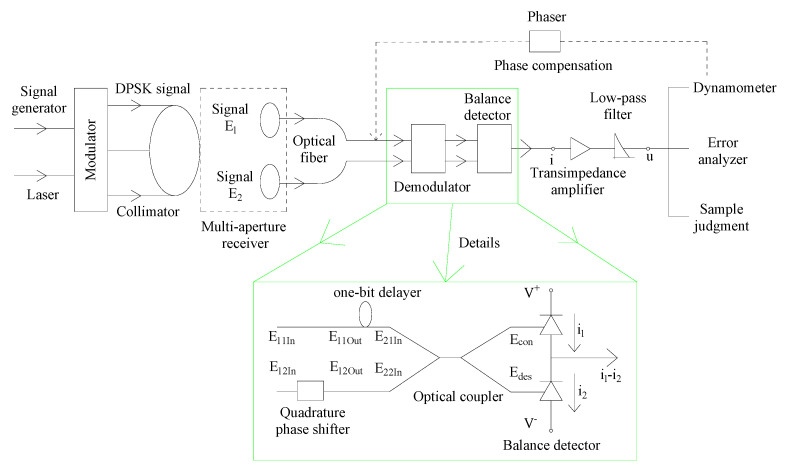
The array optical communication telescope system.

**Figure 2 micromachines-14-00409-f002:**
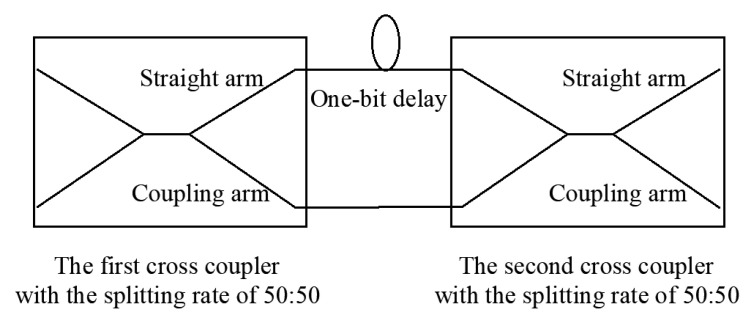
Schematic structure diagram of MZI.

**Figure 3 micromachines-14-00409-f003:**
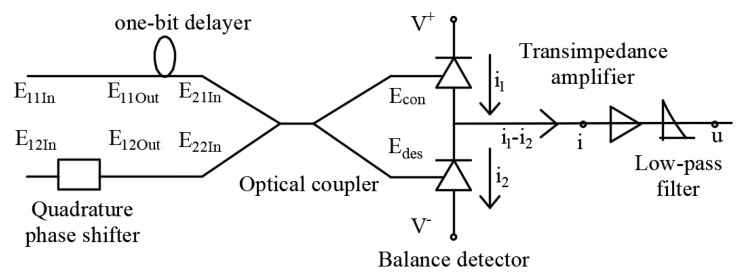
External beam splitting structure diagram of DPSK demodulation and phase detection.

**Figure 4 micromachines-14-00409-f004:**
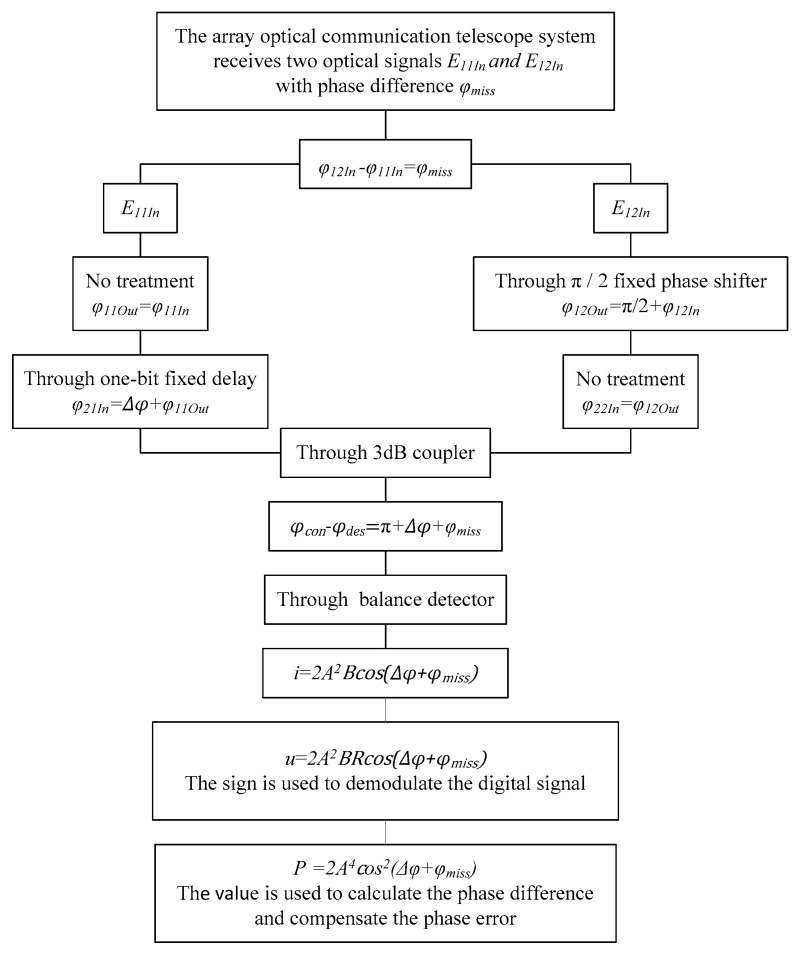
The algorithm principle of digital signal recovery and phase difference detection.

**Figure 5 micromachines-14-00409-f005:**
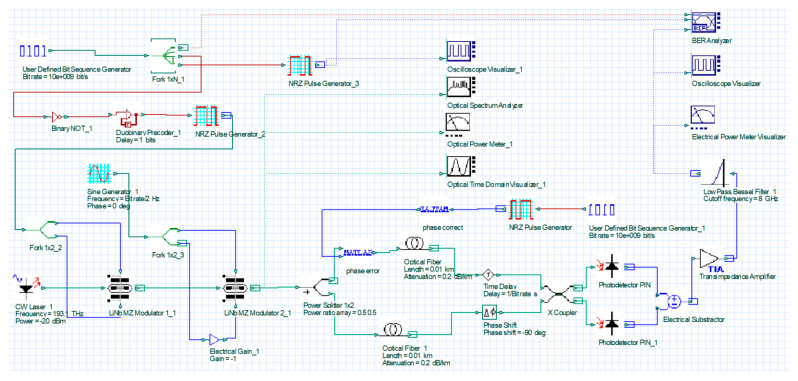
Simulation of an array optical telescope system to compensate phase using balanced detection.

**Figure 6 micromachines-14-00409-f006:**
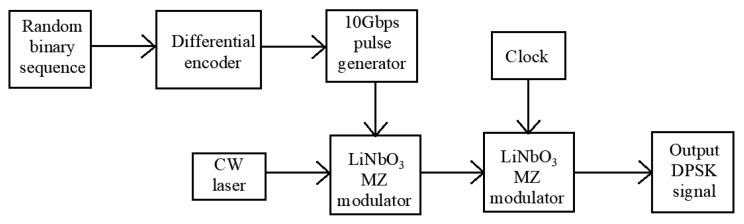
The generation process of RZ-DPSK optical signal.

**Figure 7 micromachines-14-00409-f007:**
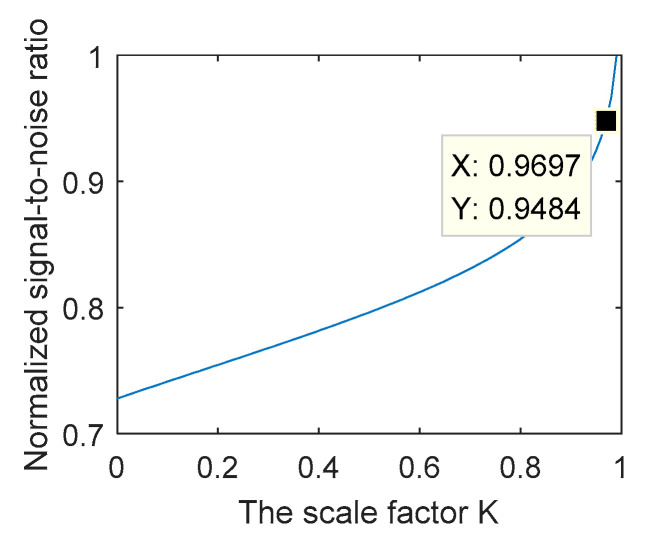
Influence of the scale coefficient on normalized signal-to-noise ratio.

**Figure 8 micromachines-14-00409-f008:**
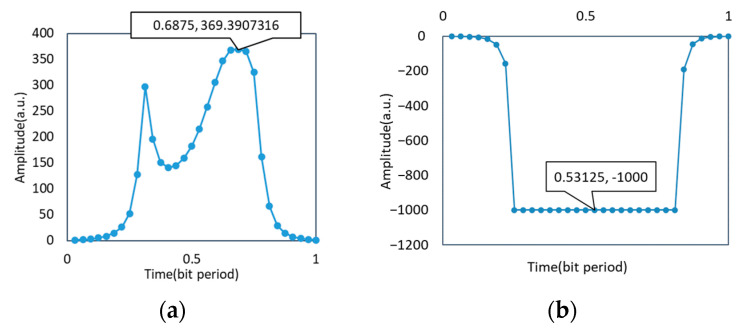
Communication quality without phase difference in the transmission of two optical signals: (**a**) Q factor without phase difference; (**b**) BER without phase difference.

**Figure 9 micromachines-14-00409-f009:**
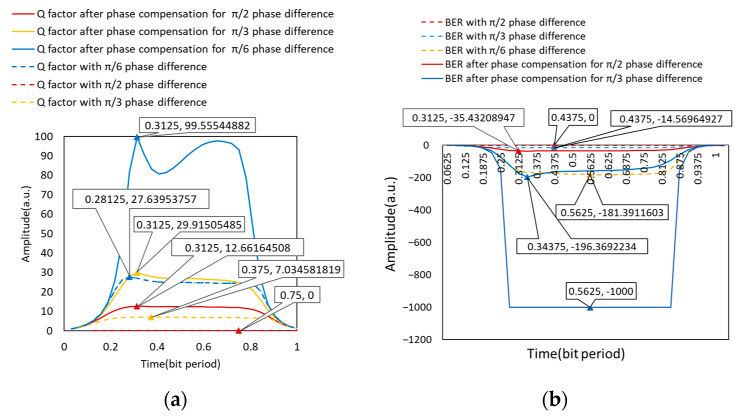
Communication quality with the phase difference in the transmission of two optical signals: (**a**) Q factors with the phase difference; (**b**) BER with the phase difference.

**Figure 10 micromachines-14-00409-f010:**
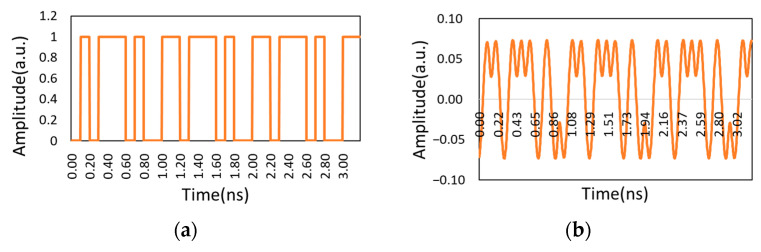
Waveform without phase difference: (**a**) Modulation waveform of CSRZ-DPSK; (**b**) Demodulation waveform without phase difference.

**Figure 11 micromachines-14-00409-f011:**
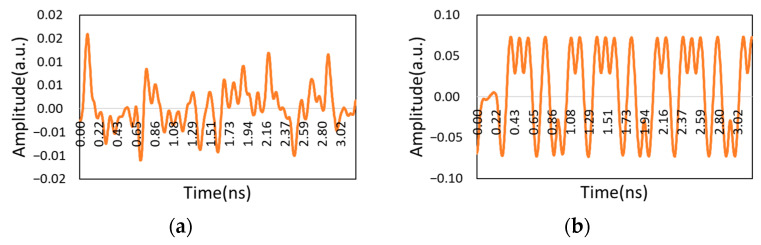
Demodulation waveform: (**a**) Demodulation waveform with π/2 phase difference; (**b**) Demodulation waveform after phase compensation.

**Figure 12 micromachines-14-00409-f012:**
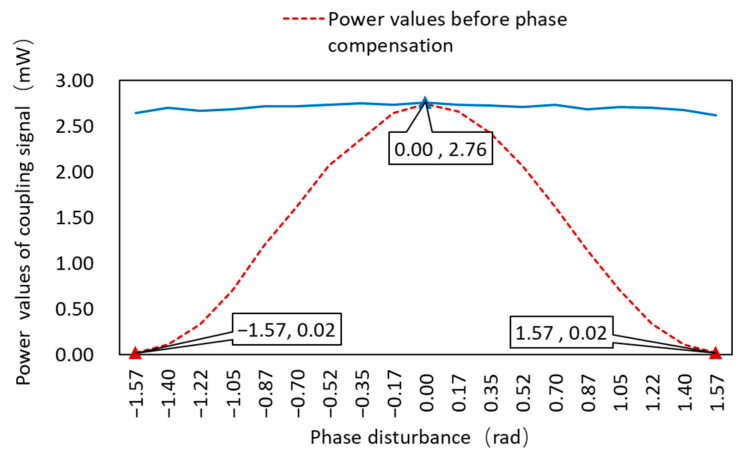
Power values before and after phase compensation.

**Figure 13 micromachines-14-00409-f013:**
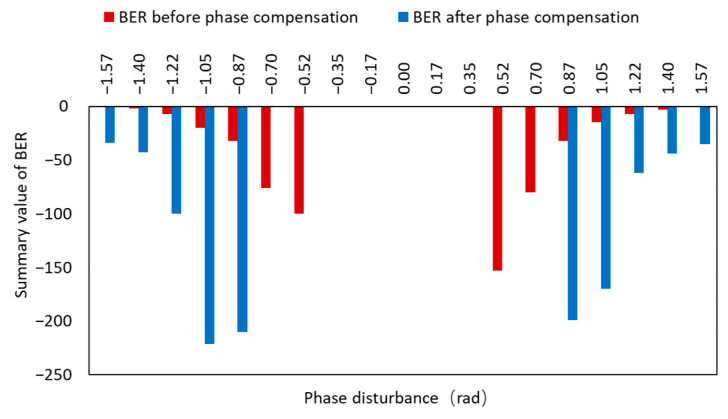
BER before and after phase compensation.

**Figure 14 micromachines-14-00409-f014:**
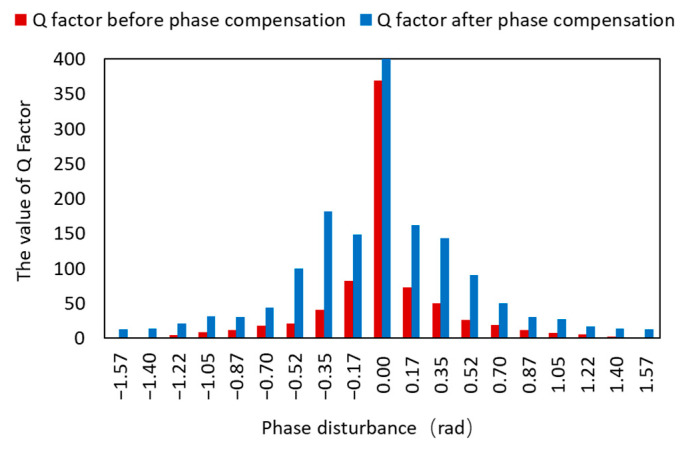
Q factors before and after phase compensation.

**Table 1 micromachines-14-00409-t001:** Generation of RZ signals with different duty cycles.

MZ Operating Points	$V_{\mathrm{bias}1}-V_{\mathrm{bias}2}$	Different Duty Cycles of RZ Signals
Peak	0	33%
Quadrature	0.5*V_π_* or 1.5*V_π_*	50%
Null	*V* * _π_ *	67%

**Table 2 micromachines-14-00409-t002:** Parameters design of a transmitter.

Component	Parameter	Value
Bit Sequence Generator	Bit rate	10 Gbit/s
NRZ Pulse Generator_3	Maximum	1 a.u.
	Minimum	0 a.u.
NRZ Pulse Generator_2	Maximum	−4 a.u.
	Minimum	0 a.u.
Sine Generator_1	Frequency	5 GHz
	Amplitude	2 a.u.
	Bias	0 a.u.
CW Laser_1	Frequency	193.1 THz
	Power	−20 dBm
	Linewidth	10 MHz
LiNb MZ Modulator 1_1	Extinction ratio	100 dB
	Switching bias voltage	4 V
	Switching RF voltage	4 V
	Bias voltage1	0 V
	Bias voltage2	0 V
LiNb MZ Modulator 2_1	Extinction ratio	50 dB
	Switching bias voltage	4 V
	Switching RF voltage	4 V
	Bias voltage1	−2 V
	Bias voltage2	2 V

**Table 3 micromachines-14-00409-t003:** Modulation pattern conversion process of DPSK signal.

PRBS	1	1	0	0	1	0	1	0	
A not gate	0	0	1	1	0	1	0	1	
A delay gate		0	0	1	1	0	1	0	1
An OR gate		0	1	0	1	1	1	1	

**Table 4 micromachines-14-00409-t004:** Parameter design of the receiver.

Component	Parameter	Value
Optical Fiber	Reference wavelength	1550 nm
	Length	0.01 km
	Attenuation	0.2 dB/km
Phase Shift	Phase	−π/2 rad
Time Delay	Delay	0.1 ns
X Couper	Coupling coefficient	0.5
	Additional loss	1 dB
PIN	Responsivity	1 A/W
TIA	Magnification	10^4^
Low-Pass Bessel Filter	Cutoff frequency	8 GHz

## Data Availability

The data are available within the article.

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
