# Peer review of "Phase Alignment of an Array Optical Telescope System Using Balanced Detection"

_micromachines, 2023, doi:10.3390/mi14020409_

Round 1

Reviewer 1 Report

The topic or this article is about the design optical telescopes to improve its detection sensitivity. The theories and design of the system is reasonable and reliable. The results are clearly presented, and the article is well organized. In my opinion, this article can be accepted as it.

Author Response

March 2023

Thank you for your kind comments and support.

Reviewer 2 Report

The authors propose a method of phase alignment of an array optical telescope using balanced detection, fundamentally, the phase difference of the signal optical path is combined with the DPSK demodulation module.

Introduction is adequate, but it could be improved with some recent advances in the subject. Sentence in line 66 “Further, different 66 from existing methods…” requires a citation. Some acronyms are well known but other are specific. For sake clarity, acronyms should be defined. Figures are appropriate, but their format and presentation should be improved (numbers on axes appear to long for example). The optical array for communication shown in Fig. 1 should be described more extensively, particularly the key issues and design challenges.

Authors claim (line 60) that they adopt DPSK modulation and multi-aperture receiving technology in order to suppress atmospheric turbulence effects, but this is not demonstrated in the manuscript (they have improved the performance, not suppressed atmospheric fluctuations effects), so, it must be phrased accordingly.

The paper is well structured and organized, however, English could be improved. There are words that are repeated frequently (line 140 and 143), or some loose phrases that requires a proper clause to connect sentences (see lines 69-70), or “It” for “They” (line 152) just to mention a few examples.

Simulation is carried out on Optisystem software. Results given in simulations do not constitute a formal proof of performance but authors could discuss in detail its reliability and drawbacks, additionally to deepen on the MATLAB joint simulation. Citations about claims of signal-to-noise ratio must be provided.

Conclusion section resembles a re-statement of the research. This section must remark the relevance of this work and its comparison with the alternative techniques, in addition to considering areas of improvement.

Reviewer 3 Report

1. In the model, the balanced detector is composed of two photodiodes with the same parameters. Actually, due to unstable fabrication process, there are differences between photodiodes, such as responsivity. The author should clarify the effect of performance differences of photodiodes on communication quality.

2. The frequency of atmospheric turbulence is about 10 kHz, to overcome the effect of atmospheric turbulence, the frequency of signal detection should at least dozens of kHz. The author should explain the speed of the whole detection system.  

3. There's an extra space between "for" and "the" in Line244.

Round 2

Reviewer 2 Report

Authors have attended most of comments and suggestions. I consider the manuscript is ready for publication.  

Figures axis numbers appear top big and overlapped inside the graphs. Maybe it is a problem with PDF viewer, but check it (just in case.)